# Immersive Virtual Reality in Stroke Rehabilitation: A Systematic Review and Meta-Analysis of Its Efficacy in Upper Limb Recovery

**DOI:** 10.3390/jcm14061783

**Published:** 2025-03-07

**Authors:** Chala Diriba Kenea, Teklu Gemechu Abessa, Dheeraj Lamba, Bruno Bonnechère

**Affiliations:** 1Department of Information Science, Faculty of Computing and Informatics, Jimma Institute of Technology, Jimma University, Jimma 378, Oromia, Ethiopia; 2REVAL Rehabilitation Research Center, Technology-Supported and Data-Driven Rehabilitation, Data Science Institute, Faculty of Rehabilitation Sciences, Hasselt University, 3590 Diepenbeek, Belgium; teklugem@yahoo.com (T.G.A.); bruno.bonnechere@uhasselt.be (B.B.); 3Department of Special Needs & Inclusive Education, Jimma University, Jimma 378, Oromia, Ethiopia; 4Department of Physiotherapy, Faculty of Medical Sciences, Institute of Health, Jimma University, Jimma 378, Oromia, Ethiopia; dheeraj.ramesh@ju.edu.et; 5Technology-Supported and Data-Driven Rehabilitation, Data Sciences Institute, Hasselt University, 3590 Diepenbeek, Belgium; 6Department of PXL—Healthcare, PXL University of Applied Sciences and Arts, 3500 Hasselt, Belgium

**Keywords:** immersive virtual reality, stroke, upper extremities, rehabilitation technology, clinical validation

## Abstract

**Background**: Immersive virtual reality (imVR) has shown promise for upper limb stroke rehabilitation (ULSR). However, optimal implementation and treatment modalities remain unclear. This systematic review and meta-analysis aimed to evaluate imVR’s efficacy in ULSR and determine optimal treatment parameters. **Methods**: A systematic review and meta-analysis of randomized controlled trials (RCTs), comparing imVR to conventional rehabilitation (CR) in adult stroke patients, was conducted. Databases including, the Web of Science, Scopus, and PubMed, were searched. Meta-regression further explored the relationship between intervention duration, frequency, and outcomes. **Results**: Twenty-three studies were included in the systematic review, representing 395 patients, with thirteen incorporated into the meta-analysis. imVR showed statistically significant improvements in the Fugl–Meyer Assessment Upper Extremity (FMA-UE) Scale (mean difference (MD) = 3.04, 95% CI [1.46; 4.62], *p* < 0.001) and the Box and Block Test (BBT) (MD = 2.85, 95% CI [0.70; 4.99], *p* = 0.009) compared to CR, but not in the Action Research Arm Test (ARAT) (MD = 3.47, 95% CI [−0.22; 7.15], *p* = 0.06). However, these improvements did not reach clinically significant thresholds (7 points for FMA-UE and 6 points for BBT). Clinical subgroup analysis showed significant improvements for both subacute (standardized mean difference (SMD) = 0.92, 95% CI [0.48; 1.36], *p* = 0.002) and chronic (SMD = 0.69, 95% CI [0.03; 1.35], *p* = 0.03) stroke stages. Meta-regression indicated that there was a significant positive relationship between the intervention duration and upper limb improvement. **Conclusions**: imVR demonstrates potential for improving upper limb motor function following stroke, particularly with longer intervention durations and individual session lengths for chronic stroke. However, the improvements observed were not clinically significant, highlighting the need for further research with larger sample sizes and standardized outcome measures to determine optimal treatment protocols.

## 1. Introduction

Stroke is the second leading cause of death and one of the most concerning global health issues [1]. It poses a significant challenge in both developed and developing countries [2]. The World Health Organization (WHO) stated that the prevalence of stroke is dramatically increasing, leading to substantial disability and mortality [3]. As highlighted in different studies, the burden of this disease is high in developing countries because of the scarcity of healthcare professionals, the limited number of rehabilitation centers, and financial constraints [4,5,6].

Eighty percent of patients with stroke experience upper limb motor impairments [7], which commonly lead to difficulty in reaching, grasping, and manipulating objects. These result in a reduction in patients’ activities of daily living (ADLs) and quality of life (QoL) [8]. Therefore, innovative and patient-centered approaches in rehabilitation are essential to enhance recovery outcomes, improve patient engagement, and address individual needs effectively [9].

Recently, innovative approaches, such as virtual reality (VR), wearable devices, mobile computing, biofeedback, augmented reality, and robotics, have dramatically enhanced rehabilitation technology [10]. Among the innovative technologies, immersive virtual reality (imVR) has recently gained more attention [11], showing promising results in improving upper motor functionality in stroke survivors [12,13]. VR can be categorized into three levels based on the level of immersion: non-immersive VR, semi-immersive VR, and imVR.

As suggested in previous studies, imVR appeared to be more effective than non-imVR and semi-imVR in upper limb stroke rehabilitation (ULSR) [14,15,16]. Furthermore, different studies have also highlighted that imVR could be more effective in helping patients recover from upper limb stroke compared to conventional therapy alone (CT) [17,18]. imVR has powerful features that provide motivation, engagement, adherence, and pain relief for patients [19,20]. Immersed in the VR environment, patients receive opportunities to recover quickly by performing more repetitions of the exercises [21].

Another powerful feature of imVR is its immersion, which allows patients to interact as if they are in a real environment. Patients may not recognize that they exist in a virtual environment [18]. This feature contributes significantly to statistically significant upper limb motor function improvements in patients performing exercise in the real world. This is enabled through the capability of imVR to produce virtual environments [22].

imVR not only has features of enjoyment and immersion, but multiple individual studies have also shown its efficacy in ULSR, increasing the functionality of the upper limb and the QoL of stroke survivors [15,23,24]. However, it is important to note that these improvements were, in most of the studies, not clinically relevant. This may be because only a few studies have specifically emphasized the role of imVR in ULSR for stroke patients and a small number of participants in experimental and control groups as well [25]. While imVR shows the potential benefits and promises of ULSR, clinicians and academics have called for more in-depth studies to explore its effectiveness [14,26,27].

There is also a clear need to further investigate the best modalities for imVR, such as the duration of interventions, settings, stroke stages (acute, subacute, and chronic), and the use of controllers or hand tracking. The duration, intensity, and frequency of intervention are key determinants of improved upper limb functionality of patients suffering from stroke [28,29]. However, the optimal treatment duration for imVR interventions in ULSR has not been comprehensively studied. This gap makes it challenging to determine the most effective treatment duration for these patients.

Additionally, it is essential to understand the optimal intervention duration for different phases (subacute and chronic) of stroke recovery. Therefore, this systematic review and meta-analysis aim to first evaluate the efficacy of imVR in ULSR; the second objective is to determine the best treatment modalities according to the different phases of the recovery process.

## 2. Materials and Methods

This systematic review and meta-analysis was conducted according to the Preferred Reporting Items for Systematic Review and Meta-Analysis (PRISMA) 2020 statement [30] and the *Cochrane Collaboration Handbook* [31]. The protocol for the review was registered in PROSPERO (CRD420246004). The PRISMA checklist is presented in Appendix A.

### 2.1. Search Strategy

The literature search was performed on three databases—the Web of Science (WoS), Scopus, and PubMed—using relevant search strings, as detailed in Table 1. The primary and review articles’ references were cross-referenced to ensure the inclusion of all relevant articles. Only peer-reviewed articles published in English were analyzed. All studies published before 10/12/2024 were considered. In total, 199 papers were retrieved, with 82 from WoS, 88 from Scopus, and 29 from PubMed.

Potentially relevant papers were imported into Zotero 6.0.26 to remove duplicates; studies were then imported and screened using the Rayyan system.

The screening process involved four authors. CK, TA, and DL screened papers based on titles and abstracts, while CK and BB conducted complete paper readings. In the end, twenty-three studies met the inclusion and exclusion criteria for systematic reviews and thirteen studies met the criteria for meta-analysis; the complete flowchart of study selection is presented in Figure 1.

### 2.2. Inclusion and Exclusion Criteria

The following PICOs criteria were used for study selection.

Population: stroke patients of all ages, severity levels, and care settings with upper limb impairments, with no restrictions based on gender.Interventions: The interventions involved using imVR to target upper limb stroke rehabilitation. There were no exclusions based on the duration of interventions, the number of sessions per week, the care settings, or the use of controllers or hand tracking.Control: conventional rehabilitationOutcomes: the primary outcome was improvement in upper limb motor function in stroke patients, evaluated using the Fugl–Meyer Assessment Upper Extremity Scale (FMA-UE), the Box and Block Test (BBT), and the Action Research Arm Test (ARAT).Study sesign: all studies included in the meta-analysis were randomized controlled trials (RCTs), whereas interventional studies were included in the systematic review section.

There were no restrictions regarding the publication date. Studies were excluded if they were non-imVR, semi-imVR, included healthy patients as a control group, included patients with comorbidities, or were published in languages other than English. The studies could be conducted in hospitals, primary or medium-level health centers, and private hospitals, irrespective of the country’s income level.

### 2.3. Data Extraction

Data extraction included parameters such as participant characteristics (age, gender, stroke type and severity), interventions characteristics (number of weeks, number of sessions per week, duration of session) and description of the imVR system (game type, game description or scenarios, setting, outcome measures), and main outcomes.

### 2.4. Quality Assessment

The Cochrane Risk of Bias 2 (RoB 2) tool was used to assess the quality of the RCTs included in the meta-analysis. The assessment was independently performed by two authors (CD and BB). Briefly, RoB2 is a comprehensive instrument used for assessing the risk of bias in RCTs. It focuses on five key domains: bias arising from the randomization process, bias due to deviations from intended interventions, bias due to missing outcome data, bias in the measurement of the outcome, and bias in the selection of the reported result.

### 2.5. Statistical Analysis

The potential effect of the imVR on UL function was examined with meta-analysis.

First, meta-analyses were performed to evaluate the effectiveness of imVR intervention using various outcome measurements, such as FMA-UE, BBT, and ARAT. A subgroup analysis was then conducted based on the stroke stage (acute, subacute, and chronic).

For studies using FMA-UE, BBT, and ARAT, the measures of the treatment effect were the mean difference (MD) and the standardized mean difference effect size (standardized mean difference (SMD)), defined as the between-group difference in mean values divided by the pooled SD computed using Hedge’s g method. These measures were used to pool results using different outcomes when comparing stroke stages.

A positive (S)MD indicates more favorable evolution in the imVR compared to the control group. As advised by the Cochrane group, we combined the results to produce a single SMD per study when several tests were used to assess the intervention in a single study [32]. Given the high heterogeneity among trials, random effect meta-analyses were performed [33].

We checked for publication bias using a funnel plot [34], and Egger’s test for the intercept was applied to check the asymmetry [35]. Sensitivity analysis was performed using the leave-one-out method to assess the robustness of the meta-analysis results. This approach involved systematically removing one study from the analysis at a time and recalculating the pooled effect estimate. This process helped identify whether any single study had a disproportionate influence on the overall results and evaluate the stability of the findings.

Statistical analyses were performed at an overall significance level of 0.05 and carried out in R Studio (4.4.1).

## 3. Results

Of the 199 research papers screened per the PRISMA guidelines, 23 were finally included in the systematic review and 13 were included for meta-analysis. The complete flowchart of study selection is presented in Figure 1.

**Figure 1 jcm-14-01783-f001:**
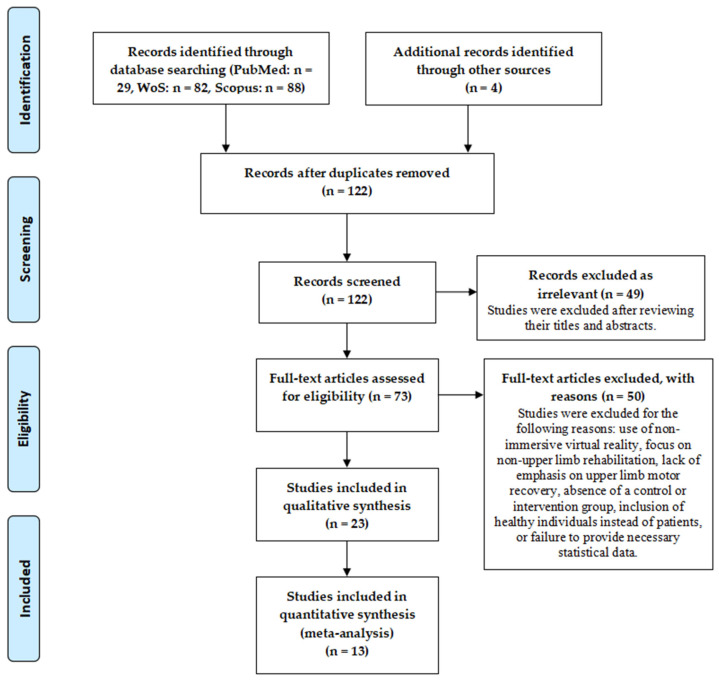
PRISMA flow diagram of study selection.

### 3.1. Quality Assessment

The RCTs evaluating the effectiveness of imVR were assessed using RoB2, as illustrated in Figure 2.

The assessment of bias arising from the randomization process indicated that 92.3% (12 out of 13 studies) exhibited a low risk, suggesting that adequate randomization procedures were generally implemented in these trials. Additionally, bias due to deviations from intended interventions was evaluated, with 69.2% (9 out of 13 studies) classified as low risk, indicating good adherence to the intervention protocols.

The risk of bias from missing outcome data was slightly more concerning. Still, 76.9% (10 out of 13 studies) were assessed as low-risk. Encouragingly, the evaluation of bias in the measurement of outcomes revealed that 92.3% (12 out of 13 studies) demonstrated a low risk, suggesting that outcome assessors were largely blinded, and the measurement processes were reliable. Moreover, bias in the selection of reported results was deemed low-risk in 69.2% (9 out of 13 studies), indicating the minimization of selective reporting.

### 3.2. Participants

In the present review, the total number of participants extracted from the relevant screened papers was 395 with 195 patients assigned to the control group. The mean age of the participants was 52 (SD = 15) years old. The ratio of males to females was 1.42:1.

The majority of the studies were performed with chronic patients (52%), followed by those for subacute patients (35%). The other studies included different stroke stages, complete information about studies and participants are presented in Table 2.

### 3.3. Interventions

The median duration of interventions for rehabilitation delivered by imVR was 3 [IQR = 3] weeks, the median duration of one single session was 30 [18] min, and the median total duration of the rehabilitation was 400 [411] min. We then analyzed these factors separately for the subacute and chronic stroke phases (Figure 3). Statistically significant differences were only found for the frequency, with higher frequency found during the subacute phase (median frequency of 5 [IQR = 2] versus 3 [1.5] sessions per week). There was no statistically significant difference for the duration of one single session (30 [4.5] versus 30 [17.5]); similarly, no difference was found in terms of total treatment duration (465 [179] versus 300 [555] min for the subacute and chronic phase, respectively). The complete details about protocols and intervention are presented in Table 3.

The intervention designs predominantly involved interactive and imVR games that encouraged patients to perform tasks mimicking real-life activities. These included reaching for targets, hitting rolling balls, grasping virtual objects like cups and balloons, and engaging in activities using a VR setting, such as dumbbell lifting, fishing, and balloon popping. Some studies explored specific techniques, like mirror therapy augmented by imVR or virtual arm illusions. Specific game-like tasks varied from study to study, such as the 360° imVR-MT game [46] and the VR-MT system [43], which involved movements like forearm rotations and tendon gliding exercises.

Regarding the technological aspect, the utilization of VR headsets was distributed as follows: Oculus Quest was the most commonly used, featuring in 8 (35%) studies, followed by Oculus Rift and HTC Vive, which both appeared in 7 (30%) studies, and Pico, which was used in a single study (5%). Most of the studies, 15 (65%), used hand-tracking interactions rather than controllers (35%) (Table 3).

### 3.4. Clinical Effectiveness of imVR

The effectiveness of imVR for ULSR was analyzed using FMA-UE, BBT, and ARAT independently. Eleven studies were included to evaluate the effectiveness of imVR for ULSR using FMA-UE [36,37,39,43,44,45,46,49,51,52,55]. The analysis showed that there is high heterogeneity among the studies included (I^2^ = 81%, τ^2^ = 3.9579), leading to the adoption of a random effects model. A statistically significant difference was found between imVR and CT (MD = 3.04, 95% CI [1.46; 4.62], *p* < 0.001).

Five studies have examined the effectiveness of imVR for ULSR using the BBT [37,40,43,46,48]. The results revealed high heterogeneity among the studies included (I^2^ = 93%, τ^2^ = 4.2491). A statistically significant difference was found in favor of imVR (MD = 2.85, 95% CI [0.70; 4.99], *p* = 0.009).

Finally, only three studies assessed the effectiveness of imVR for ULSR using ARAT [37,48,52]. These studies have high heterogeneity (I^2^ = 97%, τ^2^ = 10.1826). No statistically significant differences were found between the two interventions (MD = 3.47, 95% CI [−0.22; 7.15], *p* = 0.06); the forest plots are presented in Figure 4.

We then performed subgroup analysis to evaluate the influence of stroke severities. Due to the low number of studies assessing acute patients, we only compared the chronic and subacute phase.

Seven studies were performed with subacute stroke patients [13,39,41,42,47,52,56]. The results indicated that imVR led to a statistically significant difference in favor of imVR (SMD = 0.92, 95% CI [0.48; 1.36], *p* = 0.002).

Six studies assessed the effectiveness of imVR in chronic stroke survivors [13,38,46,50,51,54]. Again, a statistically significant difference was found in favor of imVR (SMD = 0.69 95% CI [0.03, 1.35], *p* = 0.03). No statistically significant difference was found between the chronic and subacute phase (*p* = 0.58); the forest plots are presented in Figure 5.

### 3.5. Dose–Response Relationship

The relationship between the intervention duration and the effectiveness of imVR was analyzed using meta-regression across different stroke phases (Figure 6 and Table 4). For patients in the subacute phase, the analysis revealed that the duration of a single imVR session, the frequency of sessions, and the total duration of the treatment did not significantly impact the outcomes.

In contrast, for patients in the chronic phase, the results showed a significant positive effect of the duration of a single VR session on the outcomes. The coefficient is 0.0604 (SE = 0.0251, *p* = 0.016), suggesting that longer sessions were associated with better motor function recovery. However, the frequency of sessions did not have a significant impact (β = −0.3030, SE = 0.5728, *p* = 0.59). The total treatment duration had a significant positive effect (β = 0.0024, SE = 0.0011, *p* = 0.0254), indicating that longer overall treatment durations were beneficial.

When considering the total population (both subacute and chronic phases combined), the analysis found a significant positive effect of the total treatment duration (β = 0.0013, SE = 0.0008, *p* = 0.058) and a significant positive effect of the single session duration (β = 0.0323, SE = 0.0170, *p* = 0.047). The frequency of sessions did not show a significant impact (β = −0.1148, SE = 0.2028, *p* = 0.46).

### 3.6. Risk of Bias and Sensitivity Analysis

Finally, an evaluation of potential publication bias was undertaken. The analysis of the funnel plot did not reveal significant asymmetry (Figure 7). Furthermore, the statistical assessment using Egger’s intercept yielded a value of 2.29 (SE = 1.18), with a corresponding *p*-value of 0.08. Furthermore, the sensitivity analysis (Figure 8) did not identify any study that had an extreme influence on the overall results.

## 4. Discussion

This meta-analysis investigated the efficacy of imVR for ULSR. Our findings reveal statistically significant improvements in FMA-UE (MD = 3.04, 95% CI [1.46; 4.62], *p* < 0.001) and BBT (MD = 2.85, 95% CI [0.70; 4.99], *p* = 0.009) following imVR interventions compared to conventional rehabilitation. These results align with previous research, demonstrating the positive impact of imVR on ULSR [54,57,58]. The therapeutic benefits of ImVR can be attributed to several neurophysiological mechanisms [59]. The technology’s ability to create realistic, three-dimensional environments enables the intensive practice of functional movements that directly translate to daily living activities. This immersive experience promotes neural reorganization through repeated, purposeful actions in contextually relevant settings. imVR’s effectiveness in motor rehabilitation is rooted in its comprehensive approach to sensorimotor learning [45]. The technology provides immediate, multimodal feedback that helps patients understand and adjust their movements in real time. This enhanced feedback system, combined with task-specific training in controlled virtual environments, creates optimal conditions for motor skill development. The neurological impact of imVR extends beyond basic motor practice. The technology engages multiple sensory systems simultaneously, promoting the integrated processing of visual, proprioceptive, and spatial information. This multi-sensory integration strengthens neural networks involved in movement planning and execution. Furthermore, the immersive nature of imVR activates mirror neuron pathways, which are crucial for movement observation and imitation learning [60,61]. Finally, the sense of presence and embodiment created by imVR may also enhance therapeutic outcomes by strengthening the connection between intended movements and their execution [62,63]. This heightened mind–body awareness, combined with the technology’s ability to maintain patient engagement, potentially accelerates the development and consolidation of motor skills.

However, it is crucial to acknowledge that the observed improvements did not reach clinically significant thresholds. Specifically, both FMA-UE and BBT results fell below the established minimum clinically important difference (MCID) of 7 and 6 points, respectively [64,65]. This discrepancy between statistical and clinical significance warrants careful consideration. It suggests that while imVR may lead to detectable changes, these changes may not be large enough to represent a meaningful improvement in a patient’s daily life. Future studies should consider using MCID values alongside statistical significance to provide a more comprehensive assessment of imVR’s impact. Furthermore, exploring the specific components of imVR interventions that contribute to clinically meaningful changes would be valuable.

Our subgroup analysis revealed that the effectiveness of imVR may vary across different stroke severities, though firm conclusions are limited by the small number of studies in each subgroup. For subacute stroke patients, imVR demonstrated a statistically significant improvement in upper limb motor function compared to CR. However, for chronic stroke patients, the results were less consistent, echoing the findings of previous reviews. This variability reinforces the need for future research to investigate individualized treatment approaches based on the specific needs and characteristics of patients at different stages of stroke recovery. Direct comparisons of imVR effectiveness across stroke severities in adequately powered studies are needed.

Optimizing the dosage of imVR interventions is critical for clinical practice. The studies included typically employed imVR interventions with an average duration of 3 weeks, consisting of 30 min sessions, four times per week [21,56,66,67]. However, the optimal dosage, including session duration, frequency, and overall treatment length, remains to be definitively established [68,69,70]. Our meta-regression analysis provides preliminary insights: for chronic stroke patients, longer single sessions and total treatment durations were positively associated with improved motor function recovery. These results are consistent with previous studies where longer sessions and total treatment durations were associated with better motor function recovery [71,72]. This finding is consistent with the principles of neuroplasticity and motor learning suggesting that extended exposure to imVR may be crucial for promoting long-term functional gains in this population. In contrast, these dosage parameters did not significantly influence outcomes in subacute stroke patients, perhaps reflecting the greater potential for spontaneous recovery in this earlier phase [56,73]. These findings highlight the importance of future research to determine the optimal dosage parameters for imVR interventions at different stages of stroke recovery. Such studies should consider the trade-offs between dosage, cost-effectiveness, and patient adherence [74,75].

Another important aspect to consider for clinical application and future implementation is to determine the best type of technology. imVR controller and hand-tracking technologies are two primary approaches utilizing imVR to perform virtual tasks [76]. Using controllers can be particularly challenging for patients in the acute and subacute phases of stroke recovery [77], as they may struggle to hold and manipulate the devices to perform various tasks. In contrast, hand tracking offers a more natural and immersive interaction experience than controllers. However, few studies studied and compared these methods to determine which is more suitable for specific rehabilitation goals, patient populations, or recovery phases. Thus, we are unable to analyze separately these two approaches.

This study is not without its limitations. One significant constraint is the limited sample size of studies included in the meta-analysis, which restricts the generalizability of the findings and potentially contributes to statistically non-significant outcomes. This limitation highlights the need for larger, more diverse participant groups in future research to validate the efficacy of imVR. Furthermore, concerning the quality of the study, while the RCTs included demonstrated a good level of methodological rigor, certain concerns remained regarding risks of bias, which could potentially impact the validity of our findings. Future research should focus on strategies to reduce loss to follow-up and ensure comprehensive data reporting to enhance the robustness of the evidence in terms of the effectiveness of iVR for ULSR. Additionally, the technological development of imVR games has often been for commercial rather than clinical purposes, leading to a lack of rigor in ensuring these tools are scientifically validated for specific rehabilitation stages [78]. This gap results in a scarcity of specialized applications that are tailored and validated for different phases of stroke recovery. Moreover, the reliance on controller-based interactions presents challenges for patients with acute and subacute strokes, who may struggle with the physical demands of these devices. The accurate and automatic tracking of upper limb motor movement data during exercises remains a significant limitation in the use of imVR interventions for ULSR [79]. Implementing such an approach would enable healthcare professionals and researchers to immediately assess patient progress without requiring manual data collection using outcome measurements. This would support decision-making, facilitate further analysis, and provide accurate and reliable patient data access. However, many studies have not yet incorporated technologies capable of automatically tracking upper limb movements, collecting data, and storage in a dedicated database [80]. In addition, most studies have been conducted using hand tracking, with few studies utilizing controllers [47]. This underlines the need for advancing hand-tracking technology within imVR systems to enhance accessibility and user engagement. Furthermore, most studies have been set within hospital environments, neglecting to explore the effectiveness of imVR across alternative facilities such as home settings or community clinics. This narrow scope limits the understanding of imVR’s applicability and effectiveness in diverse real-world contexts. Finally, cultural and contextual factors are rarely considered in game design, potentially diminishing patient engagement and the therapeutic value of the imVR experience [81]. Addressing these limitations is crucial for optimizing imVR interventions and expanding their implementation in stroke rehabilitation.

Despite these limitations, this study highlights the potential of imVR in ULSR. The findings indicate that while imVR can lead to statistically significant improvements in motor function, translating these enhancements into clinically meaningful outcomes remains a challenge. Clinicians, therefore, should consider incorporating imVR as a complementary tool within a more holistic rehabilitation framework rather than viewing it as a standalone solution.

One key clinical implication is the potential for imVR to augment conventional rehabilitation protocols. Its engaging, repetitive motor tasks can stimulate neuroplasticity, essential for recovery post-stroke [80]. By integrating imVR, therapists can offer more varied and motivating therapy sessions, possibly improving patient adherence and outcomes [24]. Furthermore, targeting the subacute phase of stroke recovery might yield the most significant benefits, as our subgroup analysis suggests that there is enhanced effectiveness during this period [27,82]. Clinicians should integrate imVR early in the rehabilitation process, tailoring virtual exercises to align closely with individual patient goals and functional needs.

Addressing the current gap between statistical and clinical significance, developers should collaborate closely with clinicians to design VR tasks that mirror daily life activities, thus promoting greater functional transfer from the virtual to the real world [83,84]. Advanced hand-tracking technology could further broaden imVR’s applicability, making it accessible to patients unable to manipulate traditional controllers effectively [85,86]. This feature, coupled with home-based VR applications, could play a crucial role in increasing access to intensive rehabilitation, particularly in resource-limited settings or for individuals facing mobility challenges.

Future research should therefore prioritize several key directions to address the current limitations and optimize its application and clinical implementation. Firstly, the necessity for large-scale, well-powered randomized controlled trials remains evident. Such research is vital for confirming the efficacy of imVR across diverse patient populations and settings. Determining optimal dosage parameters—encompassing intervention duration, frequency, and session length tailored to recovery stages—is another essential research focus. Moreover, incorporating cultural and contextual considerations into game design could enhance patient engagement, providing a more immersive and personalized rehabilitation experience. Secondly, there is a need to develop and rigorously validate imVR games specifically tailored to the various stages of stroke recovery. This involves collaborating with clinicians and rehabilitation specialists to ensure that these tools are both scientifically sound and clinically relevant. Advancements in hand-tracking technology should also be a focus, as they promise to enhance user accessibility and engagement, particularly for patients unable to use controllers efficiently. Expanding the scope of research beyond hospital settings to include home-based and community clinical environments will also be critical to understanding the broader applicability of imVR. This expansion can help in devising practical solutions for patients who face barriers to accessing centralized healthcare facilities. Finally, incorporating cultural and contextual elements into imVR game design will enrich the therapeutic experience, making it more engaging and effective for a global patient demographic. By addressing these future directions, imVR can be optimized as a transformative tool in the rehabilitation of stroke patients. Last but not least, another potential added value of imVR is that it could increase the access to rehabilitation services in low- and middle-income countries (LMICs) where there is a lack of rehabilitation specialist and healthcare professionals.

This is particularly alarming given the dramatically escalating prevalence of stroke in these countries. This necessitates urgent and significant investment in rehabilitation services. Without the substantial expansion and strengthening of rehabilitation services by including trained personnel, accessible facilities, and also technology-supported intervention such as imVR, many stroke survivors in LMICs will face long-term disability and reduced quality of life, placing an immense strain on families and communities [87,88]. This context makes the development of alternative rehabilitation solutions, such as imVR, particularly relevant for LMICs. ImVR technology has the potential to address these challenges by providing accessible, cost-effective rehabilitation options that can be implemented in resource-limited settings, reducing the burden on healthcare systems while maintaining therapeutic effectiveness [89,90]. However, the development and implementation of such technologies in LMICs must consider local contexts, including infrastructure limitations, cultural factors, and the need for sustainable and affordable solutions that can reach both urban and rural populations.

## 5. Conclusions

This systematic review and meta-analysis provides evidence, supporting the potential of imVR as an effective intervention for improving upper limb motor function after stroke. While statistically significant improvements were observed with imVR compared to conventional rehabilitation, particularly in FMA-UE and BBT scores, these benefits did not consistently reach clinically meaningful thresholds. Furthermore, the positive relationship between total intervention duration and functional gains, along with the benefit of longer single sessions for individuals with chronic stroke, underscores the need for the further exploration of optimal dosage parameters. Future research should prioritize larger, well-powered RCTs with standardized outcome measures and detailed reporting of intervention protocols, including game characteristics, session duration, frequency, and total treatment time. Critically, future studies should investigate the effectiveness of imVR in specific upper limb segments, explore hand tracking as an alternative to controllers, and consider culturally sensitive game development to enhance user experience and engagement. Addressing these gaps will strengthen the evidence base and contribute to the development of optimized imVR interventions for maximizing upper limb recovery following stroke.

## Figures and Tables

**Figure 2 jcm-14-01783-f002:**
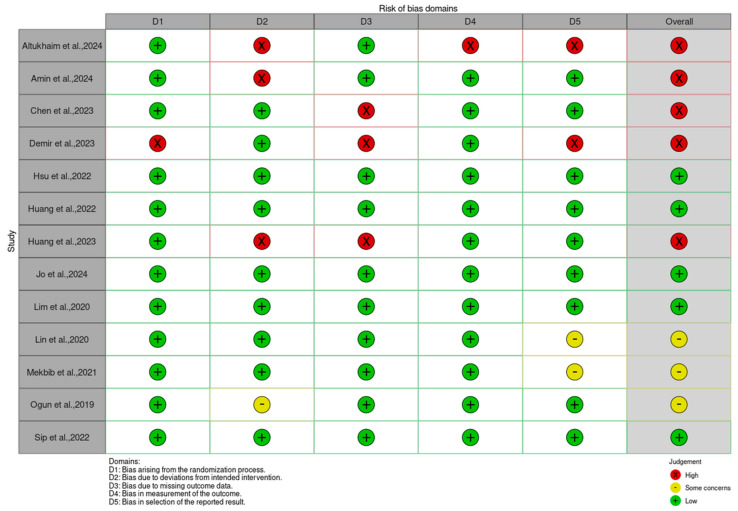
Quality assessment using the Cochrane Risk of Bias 2 tool.

**Figure 3 jcm-14-01783-f003:**
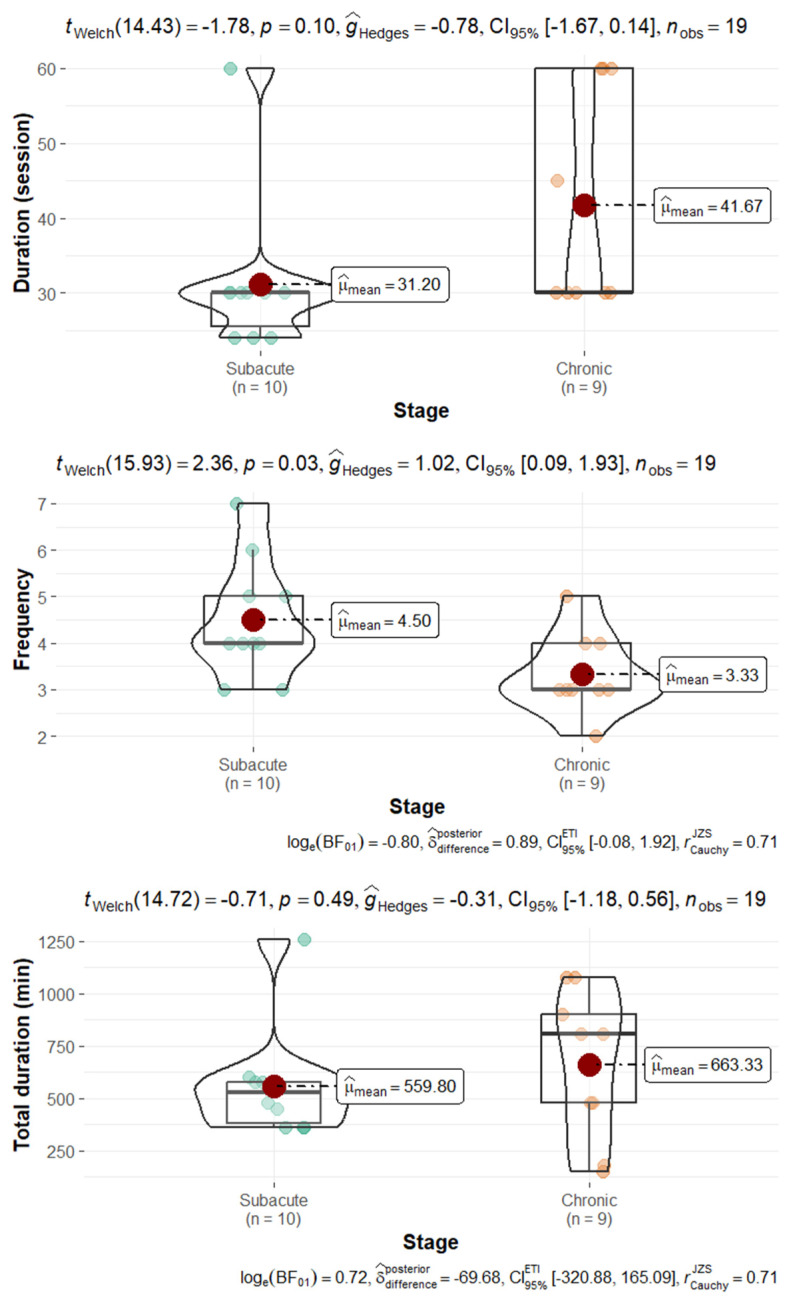
The duration of the intervention for both subacute and chronic patients.

**Figure 4 jcm-14-01783-f004:**
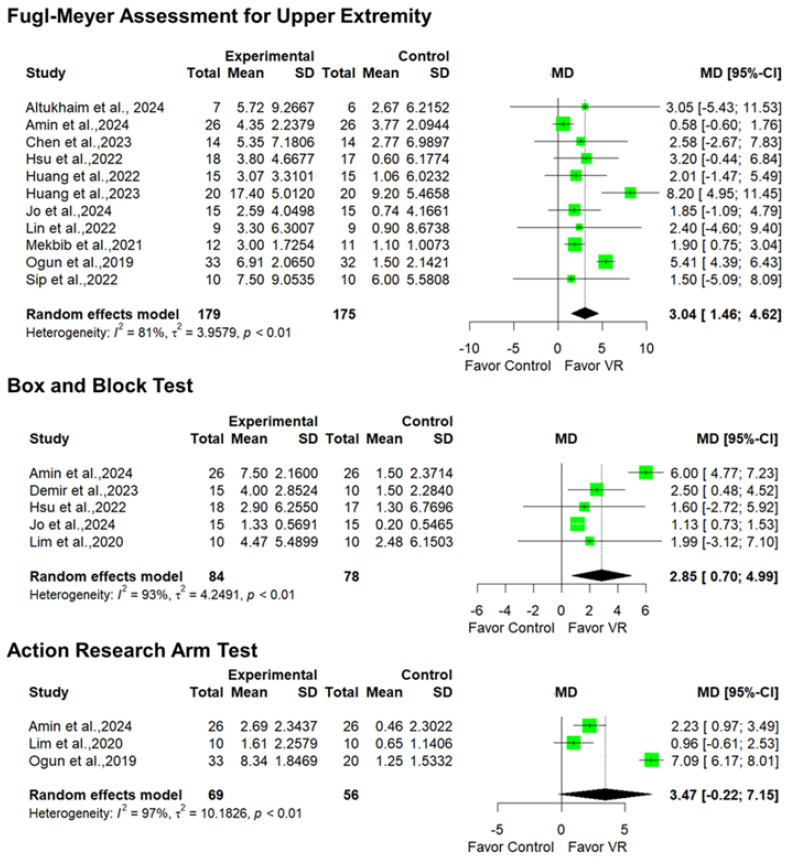
Forest plot assessing the effectiveness of imVR for ULSR with FMA-UE, BBT, and ARAT, respectively.

**Figure 5 jcm-14-01783-f005:**
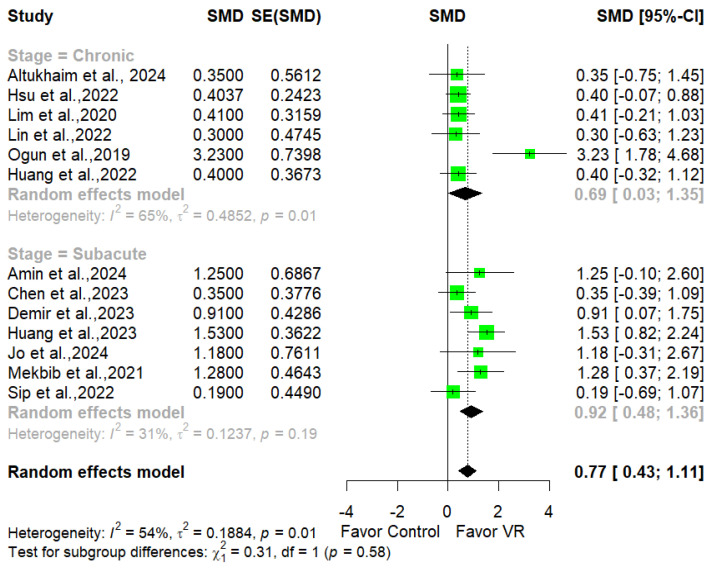
Forest plot evaluating the effectiveness of imVR in ULSR for chronic and subacute phases.

**Figure 6 jcm-14-01783-f006:**
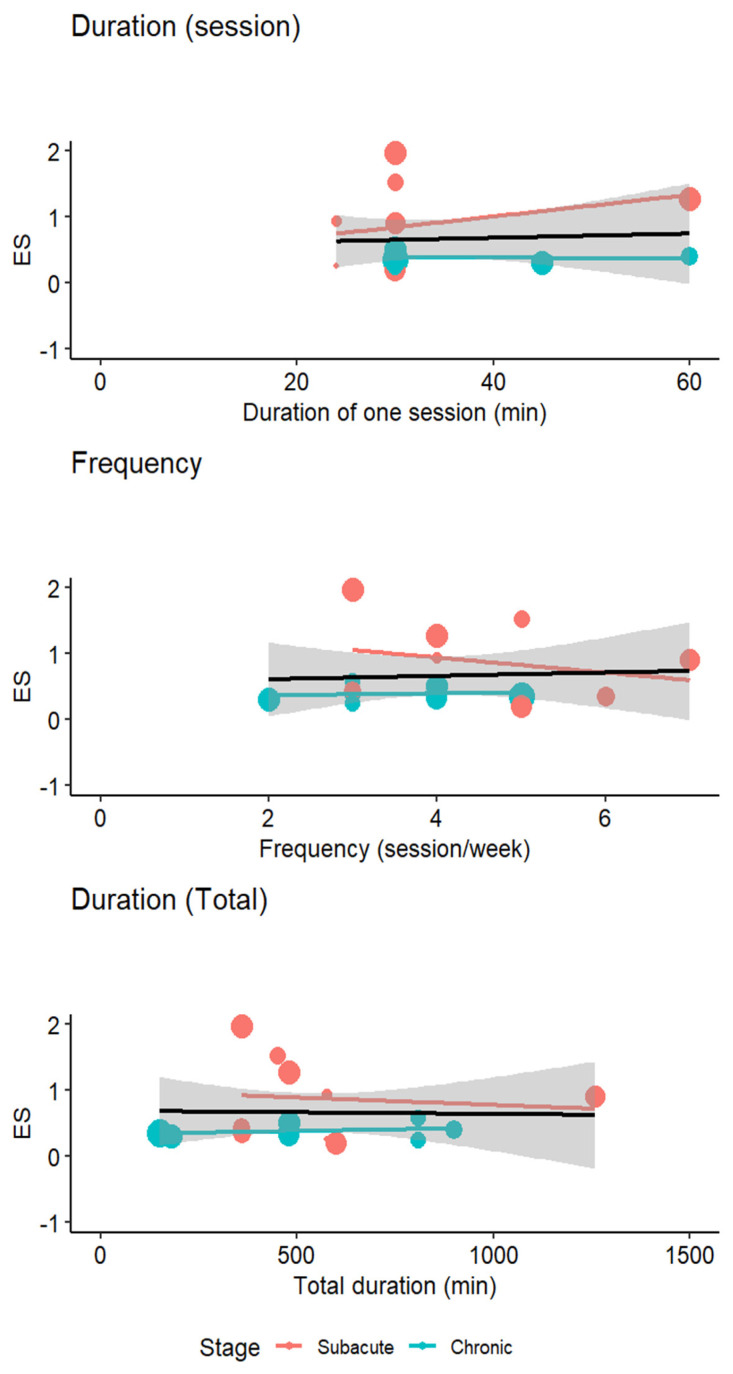
Dose–response relationship between the duration of one single session, the frequency, and the total rehabilitation duration.

**Figure 7 jcm-14-01783-f007:**
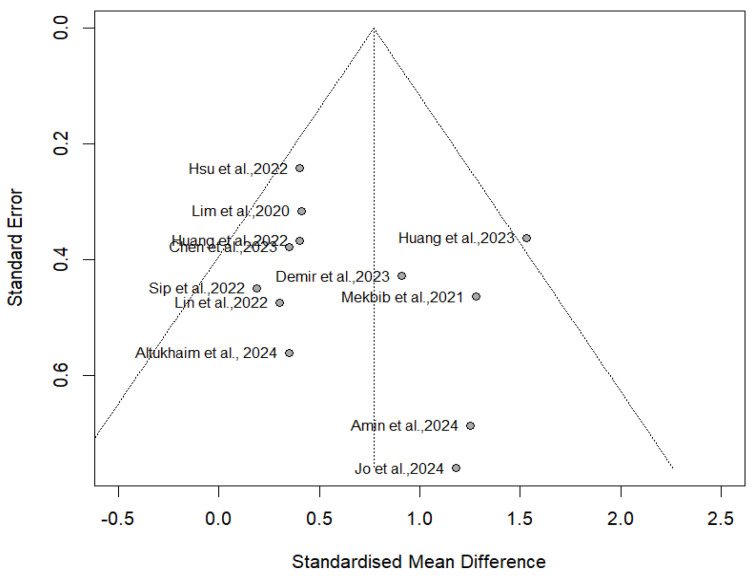
Funnel plot of publication bias.

**Figure 8 jcm-14-01783-f008:**
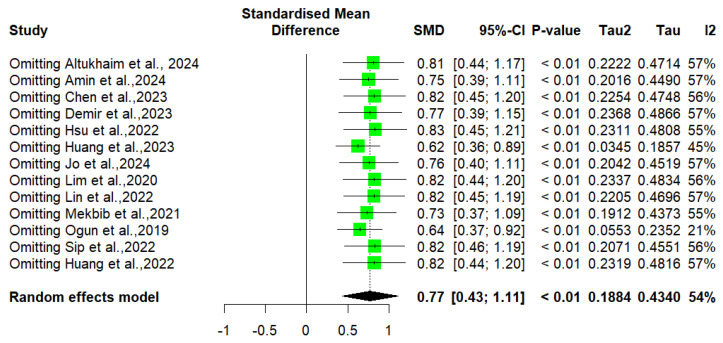
Sensitivity analysis.

**Table 1 jcm-14-01783-t001:** Systematic review sources: search databases, strings and numbers of results.

Databases	Strings	Numbers Results
Web of Science	TS = (“immersive virtual reality”) AND (“upper extremity” OR “upper limb”) AND stroke AND rehabilitat*)	82
Scopus	TITLE-ABS (“immersive virtual reality”) AND (“upper extremity” OR “upper limb”) AND stroke AND rehabilitat*)	88
PubMed	(“immersive virtual reality”[Title/Abstract]) AND (“upper extremity”[MeSH Terms]) AND (stroke[MeSH Terms]) AND (rehabilitat*[MeSH Terms])	29

**Table 2 jcm-14-01783-t002:** Characteristics of the participants.

Study	Country	N (% Female)	Control Group	Experimental Group	Age	Stroke Stage
Altukhaim et al., 2024 [36]	UK	12 (46)	6	7	72.87	Chronic
Amin et al., 2024 [37]	Pakistan	52 (35)	26	26	50.8	Subacute
Burton et al., 2022 [38]	Belgium	55 (42)	30	25	60	Acute, subacute, and chronic
Chen et al., 2023 [39]	China	28 (36)	14	14	57.75	Subacute
Demir et al., 2023 [40]	Turkey	35 (52)	10	15	51	Subacute
Elor et al., 2018 [41]	USA	6 (17)	/	6	26.5	Chronic
Everard et al., 2022 [42]	Belgium	45 (40)	23	22	64	Subacute and chronic
Fregna et al., 2022 [22]	Italy	16 (25)	/	16	62	Subacute and chronic
Hsu et al., 2022 [43]	Taiwan	35 (57)	17	18	54.6	Chronic
Huang et al., 2022 [44]	Taiwan	30 (67)	15	15	54.57	Chronic
Huang et al., 2023 [45]	Taiwan	40 (31)	20	20	64.2	Subacute
Jo et al., 2024 [46]	South Korea	30 (50)	15	15	49.43	Subacute
Juan et al., 2023 [47]	Spain	14 (36)	/	14	40.61	Chronic
Lee et al., 2020 [25]	South Korea	12 (42)	/	12	40.2	Chronic
Lim et al., 2020 [48]	Korea	20 (30)	10	10	60.25	Chronic
Lin et al., 2020 [49]	Taiwan	18 (50)	9	9	22	Chronic
Matamala-Gomez et al., 2022 [50]	Spain	20 (100)	/	20	60.05	Chronic
Mekbib et al., 2021 [51]	China	23 (26)	12	11	55	Subacute
Ogun et al., 2019 [52]	Turkey	65 (22)	32	33	60.62	Chronic
Park et al., 2021 [53]	Korea	1 (0)	/	1	56	Subacute
Phelan et al., 2021 [54]	UK	10 (60)	/	10	11	Chronic
Sip et al., 2022 [55]	Poland	20 (NS)	10	10	57	Subacute
Song and Lee, 2021 [56]	Korea	10 (40)	5	5	64	Chronic

NS: not specified.

**Table 3 jcm-14-01783-t003:** Characteristics of the individual studies included.

Study	Study Design	VR Headset	VR Interactions	Type of Exercises/Games	Description	DoI	NoSPW	DoOS (min)	Setting	Outcome Measures	Main Results
Altukhaim et al., 2024 [36]	RCT	Oculus Rift	Hand tracking	Reach the target objects	The game requires players to reach target objects in seven semi-circular positions. A total of 35 balls are presented, each target receiving five balls.	1	5	30	Hospital	FMA-UE	The study revealed that imVR has the potential to enhance motor function in stroke patients with upper limb impairment.
Amin et al., 2024 [37]	RCT	Oculus Quest 2	Hand tracking	Hit a rolling ball, grasp a balloon, switch hands, and grip a pencil.	In the first game, the patient hits randomly generated, colored rolling balls. In the second game, the patient grasps a virtual balloon to reach nearby balls. In the third game, the patient swipes incoming balls in different directions. The final game involves gripping and holding a virtual pencil.	6	4	24	Hospital	FMA-UE, ARAT, BBT	The main result showed that VR was effective in improving hand motor functions.
Chen et al., 2023 [39]	RCT	HTC Vive Pro	Controllers	Dumbbell lifting, fishing, sheep whacking, apple picking, and balloon popping.	Patients hold the controller at shoulder level for 1 to 3 s for the dumbbell exercise. In the fishing game, participants use the controller as a rod to catch fish and pull them out of the water. In the sheep game, participants stand before two holes, whacking the sheep back into the holes. In the apple-picking game, participants use the controller as a bird to pick apples from a tree and drop them onto a stump. For balloon popping, participants reach their hand toward the balloon to pop it.	2	6	30	Hospital	FMA-UE	Immersive VR statistically significant improvements in shoulder flexion, shoulder abduction, upper limb motor function, and QoL were observed in both groups.
Hsu et al., 2022 [43]	RCT	Oculus Rift	Hand tracking	VR-MT system	VR-MT included movements such as forearm supination/pronation, wrist extension/flexion, finger extension/flexion, thumb opposition with the little finger, thumb extension/flexion, and tendon-gliding exercises.	9	3	30	Hospital	FMA_UE	VR-MT has potential effects on restoring upper limb motor function in chronic stroke patients, compared to COT.
Huang et al., 2022 [44]	RCT	HTC vive	Controllers	Twenty VR exercises	There was no list of the names and descriptions of the exercises.	5	3	60	Hospital	FMA-UE	The results showed that the imVR group demonstrated significantly more improvements in FMA-UE and AROM than the COT group.
Huang et al., 2023 [45]	RCT	Oculus Rift	Controllers	Immersive VR system	NS	3	5	30	Hospital	FMA-UE, BI	The FMA-UE score was more significant in the imVR compared with the Control at the post-intervention.
Jo et al., 2024 [46]	RCT	Pico GO VR 4K	Hand tracking	Novel 360° imVR- MT	NS	4	3	30	Hospital	FMA-UE, BBT	Results revealed that the 360 imVR-MT group showed significantly more improvements in FMA-UE and BBT than conventional rehabilitation.
Lin et al., 2020 [49]	RCT	Oculus Rift	Hand tracking	Immersive VR-MT system	Supination, thumb-to-the-tip of the finger movement, thumb circling, wrist flexion and extension, tendon gliding exercise, finger flexion and extension, and key pinch.	2	2	45	Hospital	FMA-UE	The findings suggest that imVR-MT resulted in better clinical effects for upper limb motor facilitation than traditional MT.
Matamala-Gomez et al., 2022 [50]	RCT	Oculus quest	Hand tracking	Virtual arm illusion	They used exercises, organized into six modules of increasing complexity, but did not describe the exercises.	5	3	20	Hospital	FMA-UE, ROM	The imVR training group presented higher functional motor ability recovery after cast removal (T1) and six weeks later (T2) than non-imVR training groups.
Mekbib et al., 2021 [51]	RCT	HTC vive	Hand tracking	Grasping, transporting, and releasing ball	The patients pick up each ball individually and place it into a basket at the virtual table’s center.	2	4	60	Hospital	BI, FMA-UE	The VR group revealed significant improvements compared to the control group.
Ogun et al., 2019 [52]	RCT	HTC vive	Hand tracking	Types of VR programs	Cube handling, decorating a tree with leaves, picking vegetables from a bowl, kitchen experience games, and drumming.	6	3	60	Hospital	FMA-UE, ARAT	The pre-test and post-test results of the FMA-UE and ARAT showed a significant difference, favoring the VR group.
Sip et al., 2022 [55]	RCT	Oculus quest	Hand tracking	VR mirror therapy and classical mirror therapy	NS	3	6	30	Hospital	FMA-UE	FMA-UE obtained a statistically significant outcome.
Song and Lee, 2021 [56]	RCT	Oculus Rift	Controllers	Living room, kitchen, veranda, and convenience store	The content of this imVR rehabilitation game includes a daily life training component featuring environments like a living room, kitchen, and veranda.	4	5	30	Hospital	EMG and MFT	The findings indicate that imVR-based bilateral is an effective intervention for improving upper limb functions in patients with chronic stroke.
Demir et al., 2023 [40]	RCT	Oculus Rift	Hand tracking	Climb game	An immersive Oculus Rift VR climbing game was used.	6	7	30	Hospital	BBT	There were significant improvements in the imVR group compared to control group.
Lim et al., 2020 [48]	RCT	Oculus Rift	Hand tracking	Catching balls, playing xylophones, moving cherry tomatoes into a bowl, avoiding stones, throwing objects towards a target, and popping bubbles.	The patient sat in a chair with a backrest and performed six games (catching balls, playing the xylophone, moving cherry tomatoes into a bowl, avoiding stones, throwing objects) with a target and popped bubbles using both hands.	4	4	30	Hospital	BBT, ARAT	This study demonstrated that VR training combined with CR significantly improved functional improvement compared to CR alone.
Elor et al., 2018 [41]	Pilot study	HTC vive	Controllers	Catching falling stars	Patients catch descending stars that fall in a straight line (0°) in mode 1, at a 45° angle in mode 2, and 90° angle in mode 3.	1	1	5	NGO center	Questionnaire	The results suggest that an imVR intervention provides a motivating and cost-effective solution for real-time data capture during rehabilitation.
Burton et al., 2022 [38]	Observational	Oculus quest	Hand tracking	Grasp, grip, pinch and gross movement	The patients grasp and lift wooden cubes of various sizes and weights. Next, they pour water from one glass to another, grabbing and moving marbles of different diameters. Finally, they touch their neck, head, and mouth with their contralesional hand.	2	NS	NS	Hospital	ARAT, SUS	The ARAT-VR is a valid, usable, and reliable tool to improve paretic hands among individuals with stroke.
Fregna et al., 2022 [22]	Feasibility study	Oculus quest	Hand tracking	Ball in hole, cloud, glasses and rolling pin	Patients push a ball into a designated hole using their corresponding hand. Next, a cloud appears to the left or right, prompting them to pop all bubbles with the matching hand. In the third task, a glass appears on one of four pedestals arranged in a circle, and patients must push it. The final task involves using both hands to make a rolling pin a set distance along the table.	1	1	50	Hospital	FMA-UE	The results revealed that patients showed high comfort in imVR game development.
Lee et al., 2020 [25]	Feasibility Study	HTC vive	Controllers	Hammering, ball catch, cup pour, bubble touch, and playing a xylophone	The patient holds a virtual hammer to strike a nail using their affected hand, and the nail is automatically generated in virtual space. In the second activity, the patient catches a ball from the front of the virtual space and throws it back. The third activity involves pouring strawberries from a cup into a bowl. The fourth activity focuses on touching and popping a floating bubble. The final activity consists of playing a xylophone with the affected hand.	3	3	30	Hospital	ARAT	The results of the study showed significant functional improvement in all outcome measures.
Phelan et al., 2021 [54]	Feasibility study	Oculus quest	Controllers	Climbing	In this game, the child must ascend to the top by performing an overhead arm raise exercise. The game includes highlighted bricks and ropes. To climb up, the child grabs a brick and lowers their arm. Failure to grasp the brick results in the child falling off the climbing wall.	1	1	15	Hospital	ROM	Findings suggested that imVR was an engaging, enjoyable experience that distracted children from the pain and boredom of rehabilitation.
Juan et al., 2023 [47]	Comparative study	Oculus quest	Hand tracking	Lifting barbells, eating an apple and inflating a balloon.	In the first game, patients lift a barbell above a target height with their affected hand, holding it for a specified time. In the second game, they reach for an apple and bring it to their mouth, involving hand opening and closing. They also touch each finger with their thumb. The third game involves inflating a balloon to assess hand-closing ability.	NS	NS	NS	Hospital	LMS	The result of the study showed that 78.5% of the users preferred interaction using their hands.
Everard et al., 2022 [42]	Clinical trial	Oculus quest 1	Controllers	Grasping cube object	Patients move the cubes from one compartment to another.	1	1	45	Hospital	BBT	The study results revealed that test–retest reliability was excellent, and usability was nearly excellent.
Park et al., 2021 [53]	Case report	HTC vive	Hand tracking	Grasp and release	Eating, grooming, and dressing	4	5	20	Hospital	TULIA	The study reveals that an incomparably best motor response of the left hand during the imVR condition, OT, AR, and VR was 8 (26.7%), 20 (66.7%), and 28 (93.3%), respectively.

ARAT: Action Research Arm Test; BBT: Box and Block Test; BI: Barthel Index; EMG: Electromyography; FMA-UE: Fugl–Meyer Assessment for Upper Extremity; LMS: Leap Motion Sensor; MFT: Manual Function Test; NS: not specified; ROM: Range of Motion; TULIA: test of upper limb apraxia; SUS: System Usability Scale; OT: Occupational Therapy; AR: Augmented Reality; DoI: Duration of Intervention (week); NoSPW: number of sessions per/week; DoOS: Duration of One Session (min); RCT: randomized controlled trial.

**Table 4 jcm-14-01783-t004:** Meta-regressions between the effect of the VR and the outcomes for and the average duration of one rehabilitation session, the frequency (number of sessions per week), and the total amount of rehabilitation.

Condition	Duration (One Session, in min)		Frequency (Session/Week)		Duration (Total, in min)	
	β (SE)	*p*	β (SE)	*p*	β (SE)	*p*
Subacute	0.0032 (0.027)	0.90	−0.1599 (0.2200)	0.46	−0.0001 (0.0011)	0.89
Chronic	**0.0604 (0.0251)**	**0.0163**	−0.3030 (0.5728)	0.59	**0.0024 (0.0011)**	**0.0254**
TOTAL	**0.0323 (0.0170)**	**0.047**	−0.1148 (0.2028)	0.46	0.0013 (0.0008)	0.058

Bold is used to highlight statistically significant results.

## Data Availability

The dataset analyzed in the current study is available from the corresponding author on reasonable request.

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
