# Peer review of "Immersive Virtual Reality in Stroke Rehabilitation: A Systematic Review and Meta-Analysis of Its Efficacy in Upper Limb Recovery"

_jcm, 2025, doi:10.3390/jcm14061783_

Round 1

Reviewer 1 Report

Comments and Suggestions for Authors

New technologies have been emerging in the field of rehabilitation to augment the effectiveness of traditional strategies. Especially in the field of neurological rehabilitation, we have seen and we will witness a huge innovation.

The authors have focused on a very important aspect of neurorehab, the upper arms' restoration of functionality. The study of this while considering the phase of the stroke offfers a new perspective. And the data regarding the dosage are of high value.

Comments:

1. The key words used as presented in table1 are quite limited and this could lead to neglecting relevant studies.

2. Regarding the risk of bias appraisal I would suggest the use of the Cochrane Risk of Bias 2 tool. The tool used doesn't assess all the important aspects of a study.

3. Could the authors present how they addressed data variability?

4. please add to the flow chart the reasons for excluding the studies

5. in table 3 we have a summary of the included studies. It would be of high value to readers to have the results of the between groups comparison along with the p value. Thus it would be easier to see the degree of effectiveness.

6. Table 3 doesn't follow the PICOs framework, yet the authors could avoid presenting the aim (as in most studies is the same-that's why there were included in the review) and add the comparator intervention.

7. Also the manuscript lacks information on the technology used. As this is a field that rapidly evolves, it is important to see the change in the technology used.

8. the authors have thouroughly reviewed literature and this is well presented in the discussion. Please add this study and comment on their findings Mani Bharathi, V., Manimegalai, P., George, S.T. et al. A systematic review of techniques and clinical evidence to adopt virtual reality in post-stroke upper limb rehabilitation. Virtual Reality 28, 172 (2024). https://doi.org/10.1007/s10055-024-01065-1

9. The discussion is quite superficial and targets in summarizing findings. Yet, the authors don't comment on their findings in depth and present any possible causal mechanisms

Reviewer 2 Report

Comments and Suggestions for Authors

Comments to authors

The authors conducted a systematic review and meta-analysis of the effect of virtual reality on the stroke population. Despite its interest, the manuscript needs further improvement, as detailed below.

Title

  • Need to add “Effect of….” or “Efficacy of…”.

Abstract

  • “This review aimed”: Better “This systematic review and meta-analysis aimed to…”

  • FMA-UE. BBT, etc: They are not defined in the abstract (it is not known where these acronyms come from).

Introduction

  • Overall, it is well written and coherent.

  • Lines 59-61: some citations are missing.

  • Lines 90-94: should be in one paragraph.

Methods

  • Lines 97-98: Authors should declare that the PRISMA statement and the Cochrane Collaboration Handbook have been followed (and cite both).

  • In the supplementary material, the authors should include the PRISMA checklist.

  • Subsection 2.2 is unnecessary, at least in its present form. That is, all this information should be in section 2.1. In addition, in the methodology, the authors should only say what they have done, but not the results of the search. Furthermore, there is no information on whether the grey literature was searched (which should be done) and whether the references of the included studies were checked.

  • Line 119: “upper limb stroke rehabilitation 119 (ULSR) compared to”: ULSR was already defined in the introduction. Avoid redefining terms. Review manuscript for this and other acronyms.

  • “Study Design: Interventional studies”: should be RCTs.

  • The authors should redefine the exclusion criteria. Exclusion criteria are not the opposite of inclusion criteria. Exclusion criteria are those that exclude participants or studies even if they meet the inclusion criteria.

  • “The assessment was performed by the first and last authors.”: change to the initials of the authors.

  • In statistical analysis it is not really clear when to use fixed effects and when to use random effects. This should be clarified. It is also advisable to use random effects as the results are more conservative.

Results

  • Authors should provide in the supplementary material the reason for exclusion of each of the 50 excluded studies (very briefly).

  • Reference 22, which belongs to the systematic review, should not be included in the introduction. References to the review in the introduction should be avoided.

  • Figure 2: better in the supplementary material.

  • In Table 2, a column should be added with the country where the trial was conducted.

  • “such as those by [33,37,55,56] , demonstrated” This is badly written. It should say "several studies" (or something similar) and cite them. Please revise the manuscript.

  • Table 3 includes studies that are not RCTs. In fact, the first sentence of "2.3. Inclusion and exclusion criteria" states that RCTs were included. It is more important to maintain consistency than to include many studies.

  • “3.4. Clinical Effectiveness of imVR”: Although it may seem paradoxical, a p-value of 0.10 is usually used in the Egger test to indicate possible publication bias. I suggest changing this. Furthermore, it is not described in the methodology (statistical analysis section).

  • Figure 5: better in the supplementary material. Also, sensitivity analysis is not described in the statistical analysis (methodology).

  • Also, secondary analyses (sensitivity, publication bias) should be included after the meta-analysis section, not before.

  • “The total treatment duration had a mar-284 ginally significant positive effect”: whether it is marginal depends on the scale of the X-axis. If the scale of the X-axis is in minutes, there may be a relatively large increase at the end. I suggest removing the term marginal.

Discussion

  • The authors say that the improvements were not clinically significant because the improvement did not reach 7 points in FMA-UE and 6 points in BBT. However, I suggest that they clarify at least one thing for me. In the meta-analyses it is not clear whether it is SMD or MD. In the text they always say MD (Mean Difference), but in the forest plots they say SMD (Standardised Mean Difference) and on the right MD (Mean Difference). You should clarify this, because if the final results are estimated as MD, there is effectively no clinically significant improvement. However, if they are SMDs, firstly the SMDs are dimensionless units and they estimate the effect size, but without units of measurement (and it would be necessary to see whether this size is large or not). I suggest clarifying this.

  • The clinical implications should come before the limitations. That is, the limitations are the last section of the discussion (but before the conclusions).

Round 2

Reviewer 2 Report

Comments and Suggestions for Authors

Comments to authors

The authors have addressed most of the issues correctly, except for a few minor points which I detail below.

  • “This systematic review and meta-analysis was conducted according to the Preferred 99 Reporting Items for Systematic Review and Meta-Analysis Protocols (PRISMA-P) 2020”: It should be PRISMA, not PRISMA-P. PRISMA-P is for systematic review protocols.

  • If the authors have only used random effects, it is recommended that they state directly in the statistical analyses that random effects have been used. Fixed effects are more problematic and can only be used under certain conditions (few studies and low heterogeneity), but random effects can "always" be used. If the authors have finally used random effects, it is better to avoid complicating the methodology and simply indicate that random effects have been used.

  • Egger test: https://pubmed.ncbi.nlm.nih.gov/9310563/ One must rely on the original sources and their postulates.

Author Response

Thank you again for taking time to evaluate this revised manuscript and for these relevant comments.

  • “This systematic review and meta-analysis was conducted according to the Preferred 99 Reporting Items for Systematic Review and Meta-Analysis Protocols (PRISMA-P) 2020”: It should be PRISMA, not PRISMA-P. PRISMA-P is for systematic review protocols.

You are right, changes have been done.

  • If the authors have only used random effects, it is recommended that they state directly in the statistical analyses that random effects have been used. Fixed effects are more problematic and can only be used under certain conditions (few studies and low heterogeneity), but random effects can "always" be used. If the authors have finally used random effects, it is better to avoid complicating the methodology and simply indicate that random effects have been used.

Again, you are perfectly right, since we only used random effect model we modified this part of the methodology for the sake of consistency. 

  • Egger test: https://pubmed.ncbi.nlm.nih.gov/9310563/ One must rely on the original sources and their postulates.

Thank you for sharing this important resources.